# UNSUPERVISED LEARNING BASED OBJECT DETECTION USING CONTRASTIVE LEARNING

## ABSTRACT

Training image-based object detectors presents formidable challenges, as it entails not only the complexities of object detection but also the added intricacies of precisely localizing objects within potentially diverse and noisy environments. However, the collection of imagery itself can often be straightforward; for instance, cameras mounted in vehicles can effortlessly capture vast amounts of data in various real-world scenarios. In light of this, we introduce a groundbreaking method for training single-stage object detectors through unsupervised/self-supervised learning.

Our state-of-the-art approach has the potential to revolutionize the labeling process, substantially reducing the time and cost associated with manual annotation. Furthermore, it paves the way for previously unattainable research opportunities, particularly for large, diverse, and challenging datasets lacking extensive labels.

In contrast to prevalent unsupervised learning methods that primarily target classification tasks, our approach takes on the unique challenge of object detection. We pioneer the concept of intra-image contrastive learning alongside inter-image counterparts, enabling the acquisition of crucial location information essential for object detection. The method adeptly learns and represents this location information, yielding informative heatmaps. Our results showcase an outstanding accuracy of **89.2%**, marking a significant breakthrough of approximately **15x** over random initialization in the realm of unsupervised object detection within the field of computer vision.

## 1 INTRODUCTION

Object discovery is a fundamental task in computer vision, with supervised object detection making significant strides, while its unsupervised counterpart remains relatively uncharted territory Wang et al. (2022b). While large-scale labeled datasets play a pivotal role in the success of deep learning models for vision tasks Sun et al. (2017), creating such datasets is resource-intensive and time-consuming, posing limitations on their availability. This underscores the importance of reducing reliance on extensive labeled data Wang et al. (2022b).

In stark contrast to the well-explored domain of supervised object detection, unsupervised approaches have received limited attention. Moreover, most existing self-supervised learning methods have been primarily tailored for image classification tasks Wang et al. (2021) Xie et al. (2021) Wu et al. (2018), often relying on various forms of pre-training.

In this study, we embark on the journey of detecting objects within images without the need for manual annotation. Our approach draws inspiration from contrastive learning and operates in a fully class-agnostic manner, training on the COCO dataset. We identify similar objects with an impressive accuracy of 89.2%. This research explores the uncharted territory of unsupervised object detection, shedding light on its potential in the field of computer vision.

## 2 RELATED WORKS

Unsupervised object detection has long been a formidable challenge, often requiring substantial efforts to match the effectiveness of supervised learning counterparts. While recent strides have

been made in this domain, it's worth noting the unique characteristics that set our work apart from existing approaches.

Prior efforts in unsupervised object detection have explored various avenues. Some methods have relied on single images as their training data source, while others leveraged multiple images, incorporating temporal or viewpoint transformations into the mix Doersch et al. (2015) Wang & Gupta (2015) Agrawal et al. (2015). Context prediction has also been a focal point, with strategies such as predicting the relative location of a second crop in relation to the first crop or solving jigsaw puzzles Noroozi & Favaro (2016). These endeavors aimed to impart the system with an understanding of an object's constituent parts.

In recent times, visual pre-training methods have gained traction as a means to complement supervised object detection. Contrastive learning, in particular, has garnered substantial attention for its utility in unsupervised representation learning from images Noroozi & Favaro (2016) Chen et al. (2020a) He et al. (2020) Oord et al. (2018) Hénaff et al. (2019). These techniques work by mapping similar samples or different augmentations of the same instance closer together while pushing dissimilar instances farther apart, facilitating the learning process.

Additionally, some researchers have explored self-supervised methods defined as models that learn by generating its own labels Kumar et al. (2023) and weakly supervised learning methods to glean valuable visual representations Herrera et al. (2021). Notably, there has been a surge of interest in unsupervised object discovery Vo et al. (2021), Lv et al. (2023), a methodology geared toward identifying salient objects without relying on manual annotations. However, many of these approaches still hinge on the generation of masks, whether coarse or fine-grained, which effectively serve as ground truth annotations Wang et al. (2023), Wang et al. (2022a). Our method, in contrast, breaks free from the need for mask creation or annotations, offering a simpler and more straightforward approach that doesn't involve intricate training loops. This unique feature sets our work apart in the realm of unsupervised object detection.

## 2.1 CONTRIBUTIONS

We are proud to present our latest work, in which we have made the following noteworthy contributions:

- We introduce a simple, new algorithm for unsupervised object detection. Our approach, drawing inspiration from the principle of contrastive learning, seamlessly combines inter-image and intra-image contrastive techniques, thereby capturing location information for unparalleled high similarity within an image.
- We have devised a novel, modified Anchor-based NT-Xent loss function. This loss function encompasses the location information of the random crop to bolster learning. We have expanded upon the existing NT-Xent loss function to include anchor data as well.
- We achieve 89.2% accuracy on Similarity grid accuracy which is approximately 15 times greater than Random initialized grid accuracy.

## 3 OUR METHOD

### 3.1 WORKFLOW

Our approach is based on the contrastive learning method. As shown in Figure 1, for every input image ($x$), our algorithm generates two images - Image ($x_i$) which is a random crop (3.5) of the input image, and Image ($x_j$) which is an exact copy of the input image. This creates two distinct pipelines for processing the images, namely Pipeline 1 and Pipeline 2.

Pipeline 1 augments the image ($x_i$) and processes the augmented Image ($x_i$) using the ResNet architecture, and generates embeddings from the ResNet He et al. (2015) architecture. These embeddings are then passed through a projection head which reduces their dimensions. The resulting embeddings are then used as input for our loss function.

Pipeline 2 works with full-size images, specifically Image ($x_j$). It also augments the image ($x_j$) and the augmented full-size image is passed through a RetinaNetLin et al. (2017) network that

employs a Feature Pyramid Network (FPN) Lin et al. (2016) backbone built on top of a feed-forward ResNet architecture. This RetinaNetLin et al. (2017) architecture produces both a regression head and a projection (classification) head. In our experiment, we do not use regression outputs or the classification. We only use the outputs from FPN which acts as our projection head. This projection head reduces the dimensions of the embeddings, and we select the embedding that is closely linked to the embedding from Pipeline 1, i.e. a positive pair. As per He et al. (2020) and Grill et al. (2020) a positive pair is when the query and the key are data-augmented versions of the same image. The generation of positive pairs can be considered as a joint distribution over views and negative pair can be considered as a product of marginals Tian et al. (2020). We then match the embeddings based on their location information, comparing the location information from Pipeline 1 and Pipeline 2.

It is important to acknowledge that in pipeline 2, our approach trains the Feature Pyramid Network (FPN) in conjunction with the backbone i.e. Resnet.

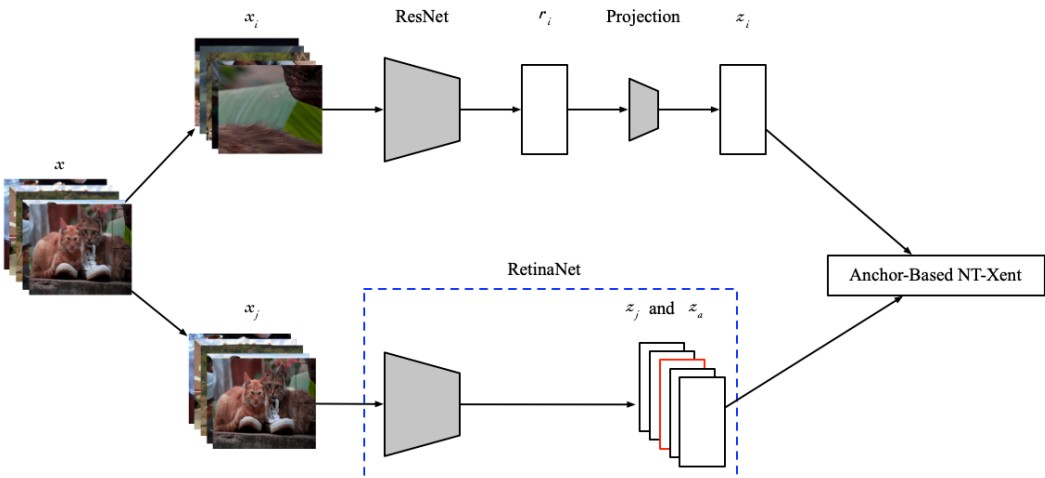

Figure 1: The diagram illustrates the interplay between our two pipelines. In the upper pipeline, refered to as Pipeline 1, we begin with input data $x_i$ and proceed to process the image, ultimately generating a representation suitable for deployment in our Anchor-Based NT-Xent loss. Similarly, the lower pipeline, referred to as Pipeline 2, takes the input $x_j$ and conducts image processing operations, culminating in the extraction of FPN outputs. These FPN outputs are thoughtfully curated to identify positive and negative samples within the image, as depicted below.

(a) In this visualization, the image on the right illustrates the utilization of FPN representations. Each black dot signifies a specific location extracted from the FPN output of a chosen layer, indicated by the red box. The red dot at the image center represents the focal point for cropping the image used in $x_i$. Within this pipeline, we identify the closest FPN location to this center, denoted as a positive counterpart, $z_j$ (white dot). Additionally, we randomly select other locations within the image, serving as anchor negatives, collectively represented as $z_a$.

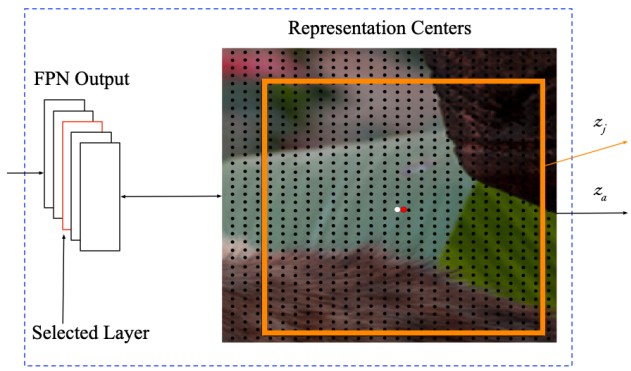

Figure 2: Selection of embeddings from FPN layers

---

**Algorithm 1** Unsupervised Object Detection

---

**Input:** Image($X$); Training epochs ($E$); Batch Size ($B$)
**for** each input image ($X$) in a minibatch **do**
    Create two images:
        Image ($x_i$) = ($x[a : a + w, b : b + h]$)
                ▷ randomly cropping image $x$ at coordinates (a,b) with width and height (w,h).
        Image ($x_j$) ≜ Image ($x$)
    Create two pipelines for processing the images:
    **Pipeline 1:**
$$x_i = \begin{cases} \text{HFlip}(x_i) & \text{with probability 0.5} \\ \text{VFlip}(x_i) & \text{with probability 0.5} \end{cases}$$
                ▷ Apply augmentations (e.g., HFlip and VFlip) to $x_i$ randomly
    $r_i = e_1(x_i) = Resnet(x_i)$
    ▷ $r_i$ is the representation generated by passing the image $x_i$ to Resnet encoder network
    $z_i = p(r_i)$
      ▷ $z_i$ is the embedding generated after passing embeddings $r_i$ in the projection head
    **Pipeline 2:**
$$x_j = \begin{cases} \text{HFlip}(x_j) & \text{with probability 0.5} \\ \text{VFlip}(x_j) & \text{with probability 0.5} \end{cases}$$
                ▷ Apply augmentations (e.g., HFlip and VFlip) to $x_i$ randomly
    $r_j = e_2(x_j) = Retinanet(x_j)$
    ▷ $r_j$ is the representation generated by passing the image $x_j$ to the RetinaNet encoder network $e_2(x_j)$
    $z_j = positive\_pair(r_j)$             ▷ Positive pair $z_j$
    $z_a = neg\_anchor(r_j)$         ▷ Randomly selected negative Anchors $z_a$
    $z_j = z_j \cdot z_a$                ▷ Embedding $z_j$
    **define** loss, $l_{i,j} = -\log \frac{\exp(\text{sim}(z_i,z_j)/\tau)}{\sum_{k=1}^{2N} 1_{k \neq i} \exp(\text{sim}(z_i,z_k)/\tau) + \sum_{k=1}^{A} \exp(\text{sim}(z_i,z_k)/\tau)}$
    Compute final loss across all positive and negative pairs
    Update the model based on the loss function's output
**end for**

---

## 3.2 OUR ALGORITHM

We propose algorithm 1 that summarizes our method. Owing to the contrastive learning approach of our method, our algorithm extends the simCLR Chen et al. (2020a) framework to learn location information as well. In the encoder network, we input a batch of pairs. Within this batch, all images except for $x_{ik}$ and $x_{jk}$ are considered negatives. Specifically, $x_{ik}$ represents a cropped image derived from the original image $x_k$, while $x_{jk}$ is an augmentation of the image $x_k$. We use two different encoder networks $e_1(\cdot)$ and $e_2(\cdot)$ i.e. one for each pipeline.

Pipeline 1 obtains the representations using the Resnet network. Hence the representations can be denoted as $r_i = e_1(x_i) = Resnet(x_i)$. For pipeline 2 the representations are obtained via the Retinanet network. This can be denoted as $r_j = e_2(x_j) = Retinanet(x_j)$. The representations $r_i$ are then passed on to the projection function represented as $p_1(\cdot)$ which yields the embeddings $z_i$. In pipeline 2, the representations $r_j$ from retinant pass throuth the FPN layers of retiant to produce embeddings $z_j$. These embeddings are mapped to our Anchor-based NT-Xent Loss function. The final loss is computed across all positive and negative pairs. It is important to note that we also sample embeddings other than the location cropped image was centered on. These embeddings are the negatives within the image known as intra-image negatives. They enable the algorithm to learn the location information within the image.

## 3.3 NEGATIVE ANCHOR SELECTION

In Figure 1, we present a visual representation elucidating the interpretation of FPN outputs within the context of our methodology. The FPN partitions the image into a grid-like structure, a fundamen-

tal component of our approach. Within the figure, one such grid level is showcased, accompanied by an image from the corresponding batch.

The black dots superimposed on the image denote individual FPN features. For the establishment of positive pairs, we utilize the center point of a bounding box (highlighted by the red dot) and perform a comparative analysis across all locations within the image grid. This procedure facilitates the identification of the grid cell closest to this center point, visually manifested as the white dot ($z_j$) in this illustrative instance.

In contrast, the procurement of negative samples involves a randomized selection process from alternative cells within the image grid. This approach ensures the availability of both positive and negative examples, thereby enhancing the efficacy of our model training process.

## 3.4 ANCHOR-BASED NT-XENT LOSS FUNCTION

We use Normalised Temperature-scaled Cross Entropy Loss (NT-Xent) Chen et al. (2020a) as our base loss function and make a few modifications to it. The original NT-Xent loss function was formed by adding a temperature parameter to N-pair loss. This temperature parameter was used to scale cosine similarities and using an appropriate parameter can help the model learn from hard negative examplesChen et al. (2020a). We expand on this knowledge in our modified loss function. While the original NT-Xent loss function has negative and positive samples, we generate anchor negative as well as positive samples. The generation of this anchor negative allows us to contrast based on the location of the crop.

$$l_{i,j} = -log \frac{exp(sim(z_i, z_j)/\tau)}{\sum_{k=1}^{2N} 1_{k \neq i} exp(sim(z_i, z_k)/\tau) + \sum_{k=1}^{A} exp(sim(z_i, z_k)/\tau)} \quad (1)$$

Where A is the set of new negative anchor and $\tau$ represents the temperature parameter Wu et al. (2018). With the introduction of anchor negatives, we are able to perform instance-level (i.e. intra-image) contrastive learning in addition to image-level (i.e. inter-image) classification contrastive learning. We also perform a few experiments limiting the number of anchor negatives as equal to batch size and as half the batch size.

### 3.4.1 INTER-IMAGE LEVEL AND INTRA-IMAGE LEVEL CONTRASTIVE LEARNING

In section 3.4, we talk about image level and instance level contrastive learning. The contrastive learning technique uses finding similar representations in the augmentations of the same input over augmentations of different inputs Saunshi et al. (2022). The popular contrastive learning approaches are generally at an image level as they cater to finding similarities between two augmented views of an image. We name such approaches as image-level or inter-image contrastive learning. We also perform contrastive learning within the same image for each instance and hence we name it as intra-image contrastive learning. The important difference between the two types of contrastive learning methods is that while Contrastive learning for image-level classification is about setting some baseline assumptions such as different images have different classes (negative-positive combinations have inherent differences at an image level) whereas With instance detection we need to pre-train the model to capture both distinct class information and location information. In order to perform an instance-level detection the model needs to learn the location information alongside the class information. Hence, we need to compare the negative pair-positive pair combination within an image as well as in addition to between images.Also, although the idea of dual contrastive learning has been applied before in the work by Li et al. (2020), to the best of our knowledge, this is the first of its kind contrastive learning approach that utilizes intra-image contrastive learning in addition to inter-image.

## 3.5 GENERATION OF CROPS:

In our network, we employ random cropping as part of our contrastive learning process. Let $W$ and $H$ represent the width and height of the input image, respectively. We generate a random crop by selecting the maximum dimension ($\max(W, H)$) and then selecting a value between 10% and 25% of this maximum dimension. This selected value becomes both the height and width of the new bounding box.

Mathematically, this can be expressed as:

$$\text{New Crop Width and Height (both)} = \text{Random}(0.10 \cdot \max(W, H), 0.25 \cdot \max(W, H))$$

The newly generated bounding box is then placed randomly within the image while ensuring that the entire crop remains within the image boundaries. This is achieved by selecting random coordinates for the top-left corner of the bounding box, denoted as $(x, y)$, such that:

$$0 \leq x \leq W - \text{New Crop Width}$$
$$0 \leq y \leq H - \text{New Crop Height}$$

This guarantees that the crop is always contained within the image, and no edges of the crop extend beyond the image's boundaries. We perform batch-based image padding, where we determine the maximum width and height within each batch. Subsequently, we pad the images to match the required size based on this maximum width and height.

### 3.6 AUGMENTATIONS

Data augmentations have been extensively researched and validated as a powerful technique in various vision tasks Simonyan & Zisserman (2014) Liu et al. (2015) Qi et al. (2019) Ratner et al. (2017) He et al. (2020) Shorten & Khoshgoftaar (2019) and have become a crucial element in achieving state-of-the-art results. In this experiment, we have also studied the impact of augmentations on our proposed method. We have used various augmentations such as Horizontal Flip (HFlip), Vertical Flip (VFlip) (Figure. 3) and applied them to the image in random order. We have avoided incorporating augmentations that cause significant color distortion, as they have been shown to decrease the overall gains He et al. (2020).

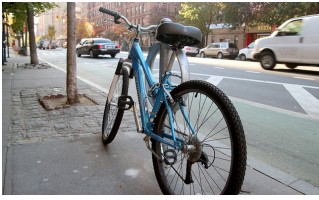
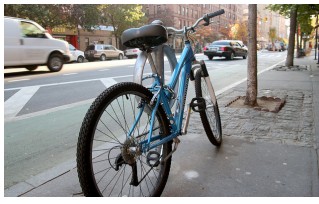
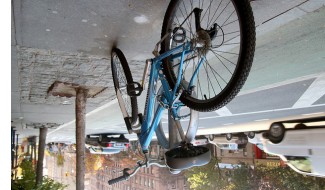

| (a) Original Image | (b) Horizontal Flip | (c) Vertical Flip |

Figure 3: Augmentations

## 4 EXPERIMENTS

Our approach is exclusively trained on images from MS-COCO (Lin et al. (2014)) and is evaluated in a zero-shot manner, without any fine-tuning of labels or data. The experiments aim to identify the regions of highest similarity within an image when compared to a cropped image.

It is a well-known fact that contrastive learning methods tend to necessitate substantial memory banks and can be relatively time-consuming to train (Chen et al. (2020a);He et al. (2020);Chen et al. (2020b); Caron et al. (2020);Xie et al. (2021)). As such, we elected to limit our model training to 200 epochs in this study. Despite this, we found that these iterations were sufficient to discern the trends our model was generating. In the future, we intend to expand our experiments further, taking into account the necessary time and computational resources, so that more comprehensive evaluations can be carried out.

In our experimental setup, we conducted end-to-end training of our models. As previously mentioned, our model adopts a non-siamese architecture composed of two distinct pipelines. We trained the model utilizing the Adam Kingma & Ba (2015) optimizer, and for the backbone of both pipelines, we employed the ResNet18 architecture. Due to the image size requirements, we employed a batch size of 4, as the images necessitated a minimum size of 608 pixels, resulting in high memory usage. To generate negatives within each image, we chose 10 anchor negatives per image, resulting in a total of 40 aditional negative locations in each batch.

## 4.1 HEATMAPS

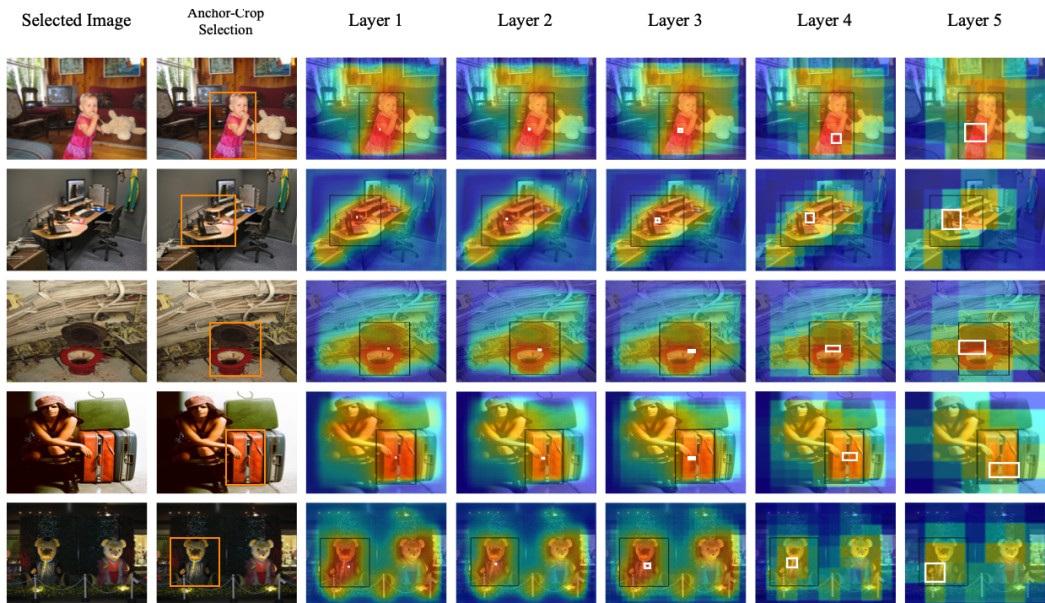

Figure 4: In this visual representation, we present the model's output per layer. Our approach involves showcasing the representations obtained at each layer of the Feature Pyramid Network (FPN) and distributing the corresponding similarities, centered around the FPN grid cells, across the entire image. This approach enables the generation of heatmaps for each layer of the FPN, providing valuable insights into the model's hierarchical feature representations.

In these figures, we employ a colormap to render our heatmaps, which span the gradient from blue to red. In this color scheme, blue signifies areas of low similarity, whereas red indicates regions of high similarity. We have created two distinct figures to elucidate and discuss the output of our model.

Figure 4, we can observe the presence of an orange bounding box, which signifies the random crop generated from the original image. This crop serves as a focal point for our analysis. Additionally, within each heatmap, a white bounding box is displayed, indicating the center of the grid with the maximum similarity. This distinctive feature aids in pinpointing the areas of greatest interest within the generated heatmaps, providing a comprehensive perspective on the patterns and correlations identified by our model.

Furthermore, in Figure 4, we focus on highlighting the disparity between the layers of the FPN. This examination provides valuable insights into the hierarchical representation of features within our model. Meanwhile, in Figure 5, our emphasis shifts towards illustrating the dynamic behavior that emerges from the aggregation of these FPN layers, culminating in the production of a heatmap reflecting the combined similarity across different levels. The utilization of this combined output serves as a critical step in our process for selecting the highest similarity score images within the entire dataset.

In Figure 5, we present a comprehensive analysis of how our model meticulously selects images sharing common attributes. For example, in the first row, we observe the model's inclination towards images featuring crowds in the background, showcasing its ability to effectively discern and group such content. In the second row, it becomes apparent that the model excels at identifying images containing water or snow-related elements, underscoring its proficiency in recognizing specific environmental characteristics. Lastly, the third row highlights instances where the model excels in identifying images primarily focused on subjects related to food. This in-depth analysis underscores the model's robustness in capturing and categorizing a diverse range of visual content, significantly enhancing our understanding of its performance.

Figure 5: This figure shows a grid of images gathered after selecting a crop within the dataset and searching the top10 similar images. The selected crop is passed to the RetinaNet to produce a representation and the highest similarity images are process on a batch to produce the FPN outputs used to compare and execute the selection.

## 5 RESULTS

We introduce the following three key metrics to evaluate grid alignment performance:

**Definition 1: Similarity Grid Accuracy (SGA)** SGA measures the accuracy of the similarity grid's alignment with the bounding box within a dataset of images.

$$SGA = \frac{SGI}{N} \times 100\%\qquad(2)$$

Where:

$SGA$ is the Similarity Grid Accuracy.

$N$ is the total number of images in the dataset.

$SGI$ is the count of instances where the similarity grid is entirely contained within the bounding box for a given image.

**Definition 2: Random Initialization Grid Accuracy (RIGA)** RIGA evaluates the accuracy of randomly initialized grids in terms of their alignment with bounding boxes within the same dataset of images.

$$RIGA = \frac{RIGI}{N} \times 100\%\qquad(3)$$

Where:

$RIGA$ is the Random Initialization Grid Accuracy.

$N$ is the total number of images in the dataset.

$RIGI$ is the count of instances where a randomly initialized grid is entirely contained within the bounding box for a given ima

Table 1: Accuracy per layer of FPN

| *Layer* | *SGA* | *RIGA* | *GAP-R* |
|---------|-------|--------|---------|
| Layer 0 | 0.892 | 0.0601 | 14.882  |
| Layer 1 | 0.889 | 0.0601 | 14.792  |
| Layer 2 | 0.881 | 0.0601 | 14.659  |
| Layer 3 | 0.848 | 0.0610 | 13.902  |
| Layer 4 | 0.709 | 0.0614 | 11.547  |

**Definition 3: Grid Alignment Performance Ratio (GAP-R)** GAP-R quantifies the alignment performance of the similarity grid relative to random initialization within the dataset.

$$GAP\text{-}R = \frac{SGA}{RIGA} \tag{4}$$

A GAP-R value greater than 1 indicates superior alignment of the similarity grid with the bounding box compared to random initialization, while a value less than 1 suggests the opposite. This metric provides a straightforward method to assess grid alignment accuracy across different initialization methods.

In Table 1, we present the layer-by-layer (of FPN) Similarity Grid Accuracy (SGA), Random Initialization Grid Accuracy (RIGA), as well as the Grid Alignment Performance Ratio (GAP-R). All results are based on the evaluation of COCO-5k images.

We observe that using our algorithm, the highest similarity grid aligns inside the bounding box 89.2% of the time, compared to only 6% of the time with random initialization. Our method also outperforms random initialization by a factor of approximately 14.882, as denoted by *RIGA*.

## 6 CONCLUSION

In this study, our goal was to simplify the arduous process of labeling for object detection applications. We built upon the foundation of visual pre-training, striving to take it a step further by entirely replacing the need for visual pre-training. In essence, our research delves into the implications of relying solely on unsupervised learning for detection tasks.

To achieve this, we harnessed feature learning techniques akin to those employed in widely used supervised learning approaches but adapted them to an unsupervised learning framework. This adaptation facilitated the localization of objects of interest and enabled us to visualize their closest counterparts with remarkable accuracy. Hence, our method has the potential to revolutionize the labeling process, substantially reducing the time and cost associated with manual annotation.

Our results speak to the efficacy of our method, as we achieved an impressive 89.2% accuracy in identifying similarities to the object of interest. This represents a substantial improvement, nearly 15 times better than the results obtained without the application of our method.

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
