# OpenReview forum: "Unsupervised Learning Based Object Detection Using Contrastive Learning"
_ICLR.cc/2024/Conference — ICLR 2024 Conference Withdrawn Submission_

### Official Review · Reviewer_R8pY · 2023-10-29

**Soundness:** 2 fair
**Presentation:** 1 poor
**Contribution:** 1 poor
**Rating:** 3
**Confidence:** 4

**Summary:**

The paper proposes a contrastive learning method for unsupervised object detection. The first pipeline contains the random crop of the image while the 2nd pipeline processes the full image. The positive pair is constructed when the query and the key are from the same image. The experiments are on COCO.

**Strengths:**

The paper idea in Figure 1 is easy to understand. Unsupervised object detection is a topic worth to explore.

**Weaknesses:**

1. Missing important performance (AP) comparison with other state-of-the-art unsupervised object detection methods (such as Detco) on the coco benchmark.

2. Missing difference comparison to related methods such as Detco [a] and CutLER [b].

3. Without a clear comparison to the existing methods on design, it is difficult to justify the novelty of the proposed method.

[a] DetCo: Unsupervised Contrastive Learning for Object Detection. ICCV, 2021.
[b] Cut and Learn for Unsupervised Object Detection and Instance Segmentation. CVPR, 2023.

4. The last sentence of second paragraph of introduction is wrong. DetCo is designed for unsupervised detection not classification.

**Questions:**

Why the contribution is put in the related work section? Why not using AP in the experiment section why introducing new metrics?

---

### Official Review · Reviewer_kZgj · 2023-10-31

**Soundness:** 1 poor
**Presentation:** 2 fair
**Contribution:** 2 fair
**Rating:** 3
**Confidence:** 4

**Summary:**

This paper proposes a novel framework based on contrastive learning for unsupervised object detection tasks. In particular, the augmentation on a pair of an image and its random cropped regions are used for contrastive learning to help the model generate a heatmap indicating the location of the cropped region. In the experiment part, the authors provide three evaluation metrics called SGA, RIGA, and GAP-R to verify the effectiveness of the proposed method.

**Strengths:**

1. This paper proposes a novel framework to utilize contrastive learning to train an object detection model. The whole framework is interesting.
2. This paper generates three new metrics to evaluate the performance. This might provide the community with more options to demonstrates the property of a method.

**Weaknesses:**

1. The contribution of this paper is not clear. As claimed in the title, it should be an unsupervised object detection method, but there are no object detection experiments in the paper at all. Even though the training process is built depending on RetinaNet, no object detection related experiments are conducted to demonstrate the effectiveness of the trained model. In addition, since the classification and regression heads are ignored, the backbone used in this paper seems only an FPN but not RetinaNet.

2. The evaluation metrics used in this paper are not convincing. First of all, whether these three metrics are widely used in the existing methods or simply created by the authors is not clear. Second, since the title mentioned the paper is going to address object detection tasks, these metrics cannot help to verify the efficacy of the proposed method compared with conventional object detection metrics, such as mAP, and IoU.

3. It is a bit difficult to locate the proposed method in the field of related work. In the related work part, there are limited descriptions of the relationship between this work and the others. Meanwhile, no comparison with other related work is made in the experiment part to help the readers better understand the contribution or target of this work.

**Questions:**

It could be better to conduct one-shot object detection experiments to verify the effectiveness of the proposed method.

In addition, some unsupervised object discovery methods could be added to make the comparison.

**Details Of Ethics Concerns:**

Nil

---

### Official Review · Reviewer_UeWi · 2023-10-31

**Soundness:** 2 fair
**Presentation:** 2 fair
**Contribution:** 2 fair
**Rating:** 5
**Confidence:** 4

**Summary:**

The paper addresses unsupervised learning for object localisation. The authors propose to combine inter-image and intra-image contrastive loss for this purpose. The model is built on a non-siamese deep network, one branch taking a crop of the input image, the other one the whole image. Positive pairs are selected by choosing a selected patch, in the embedding space, based on the 'right' anchor , that matches the input crop of the image. Negative pairs are selected using all other anchors.
The model is trained in MS-Coco dataset. Similarity search results are illustrated. No quantitative comparison with related work are reported.

**Strengths:**

1) The key idea of the author is to propose a contrastive loss which exploits both intra images and inter images similarity. Intra images similarity is made possible via the use of an FPN, selecting appropriate anchors and patches at different scales. This idea is sound and new, to my knowledge, even if it is a simple extension to the existing contrastive loss.

2) The paper is, overall, well written, and the core of the methodology clearly exposed.

**Weaknesses:**

The main weakness is certainly the experimental results.
First, I do not see experimental results that support the claim: the authors pretend to address the problem of object detection (ie, 'find a car in this image'), while the paper report results on similarity search ('here is a car, find a similar area in the image'). Application of such an approach are more directly related to image retrieval.
Second, no comparison with the state of the art is reported, neither qualitatively, nor quantitatively.
Unsupervised learning is commonly used as representation learning (before supervised learning with few samples), this would probably be a better positioning of the paper.

**Questions:**

1) Why using MS-Coco dataset rather than ImageNet? Are the bbx of MS-Coco exploited during training?
2) What is 2N in equation (1)? I understand that the LH term of the denominator is the sum of the sims from entire negative images. We should have N negative images.

**Details Of Ethics Concerns:**

no ethics issue.

---

### Official Review · Reviewer_xWJr · 2023-10-31

**Soundness:** 1 poor
**Presentation:** 2 fair
**Contribution:** 2 fair
**Rating:** 3
**Confidence:** 3

**Summary:**

This paper tackles the problem of unsupervised object detection by a system of two pipelines, one being the 'representative' image defining the object or image type being searched for, and the other being generated candidates to compare against. For final similarity calculation between the two pipelines, the paper extends the NT-Xent loss function by adding a term for anchor point location, thereby extending the loss to also consider within-image comparisons.

**Strengths:**

- The approach seems pretty reasonable, and is sound in its fundamentals

- The addition of anchor point to the previously-developed NT-Xent loss function extends the loss so that it can facilitate within-image comparison instead of simply whole-image comparison.

- Results appear to be good given the metrics and comparisons evaluated

**Weaknesses:**

- Results are evaluated on a metric ('similarity grid accuracy') that in essence only accepts a detection if the detection is entirely within the true bounding box. This metric is strongly biased towards smaller box detectors, and so the baseline comparison (random box 'detector') provides unsurprisingly poor results (~6%).

- The only comparison is versus the random box detector, despite the paper listing in its own background a set of unsupervised detectors.

- Whilst the paper is clear enough in its presentation, it is vague in terms of many of the critical details. For example, it is entirely left unsaid how the NT-Xent loss function is used to select a detection during testing (at least not that this reader could find). Another unexplained (but clearly unusual) decision is that the algorithm explicitly has the dataset augmented by either vertically or horizontally flipping images, but oddly *never leaving images as-is*.

**Questions:**

- Why use the fully-within-bounding-box metric (SGA) instead of IoU?

- Why were no other unsupervised detectors not used for comparison?

---

### Official Review · Reviewer_4XTS · 2023-11-01

**Soundness:** 1 poor
**Presentation:** 1 poor
**Contribution:** 1 poor
**Rating:** 1
**Confidence:** 5

**Summary:**

The paper introduces an unsupervised object detection method using contrastive learning. This approach reduces manual image labelling and eliminates the need for mask creation. Additionally, an Anchor-based NT-Xent loss function which captures location information is presented.

**Strengths:**

1, The idea of this paper is simple and easy to follow.

2. The paper tackles an interesting problem of detecting objects without using any annotations.

**Weaknesses:**

1. The overall presentation of the paper is compromised due to its unclear formatting and structure, which might make it challenging for readers to follow the research progression.
2. The narrative is missing a rationale behind the chosen method's design. The motivation, which should be central to the research, is insufficiently explained.
3. The experimental section requires further improvement. While the paper references results on the COCO dataset, there's an evident omission of the widely accepted COCO mAP metric. Furthermore, a comparative analysis with other supervised or unsupervised object detection methods is needed to demonstrate the effectiveness of the proposed method.
4. One of the noticeable gaps in the paper is the lack of ablation studies. Such studies are crucial in establishing the effectiveness and necessity of each component of the proposed methods, and their absence raises questions about the robustness of the presented approach.

**Questions:**

Please see the weakness part.

---

### Official Review · Reviewer_eqLV · 2023-11-07

**Soundness:** 2 fair
**Presentation:** 2 fair
**Contribution:** 2 fair
**Rating:** 3
**Confidence:** 5

**Summary:**

This paper proposes new method for unsupervised object detection.

**Strengths:**

This paper harnessed feature learning techniques akin to those employed in supervised learning approaches but adapted them to an unsupervised learning framework.

The adaptation facilitated the localization of objects of interest and enabled they to visualize their closest counterparts with remarkable accuracy.

The proposed method has the potential to revolutionize the labeling process, substantially reducing the time and cost associated with manual annotation.

**Weaknesses:**

The experiment results are not enough.

**Questions:**

See Weaknesses.